# Analysis of the Mechanical Degradability of Biodegradable Polymer-Based Bags in Different Environments

Martina Fileš [1], Anja Ludaš [2], Sanja Ercegović Ražić [2,*] and Sandra Hudina [1]

1. Department of Biology, Faculty of Science, University of Zagreb, Horvatovac 102a, 10000 Zagreb, Croatia; files.martina@gmail.com (M.F.); sandra.hudina@biol.pmf.hr (S.H.)
2. Department of Materials, Fibers and Textile Testing, Faculty of Textile Technology, University of Zagreb, Prilaz baruna Filipovića 28a, 10000 Zagreb, Croatia; anja.ludas@ttf.unizg.hr
* Correspondence: sanja.ercegovic@ttf.unizg.hr

**Abstract:** Biodegradable polymer-based bags were developed as an alternative to plastic. However, their degradation in environmental conditions has not been fully investigated and is often incomplete. Here, the decomposition of three types of biodegradable bags and one type of plastic bag in different types of environments was analyzed. Polymer bags were exposed for six weeks in water, soil, air and compost, while the control groups were stored in room conditions. All types of polymer bags were sampled twice (after 3 and 6 weeks), and different parameters of changes in physical–mechanical properties were measured. The research established significant differences in changes in mechanical properties between different types of biodegradable polymer bags, with 'white' and 'brown' bags showing the best decomposition potential. As expected, the largest change in the structure and physical–mechanical properties of all types of polymer bags was recorded in compost, and the smallest in air and water.

**Keywords:** environment; biodegradable polymers; physical–mechanical properties; damage of surface structure; waste

## 1. Introduction

Today, plastic in general is an irreplaceable and ubiquitous polymer material that is widely and importantly used in the production of packaging, transportation, construction, medicine, etc. Its advantages are that it is widely used. Its advantages are also that it is widely applicable, affordable, lightweight and at the same time durable. The advantages and disadvantages of plastic have never been more apparent than during the COVID-19 pandemic, where plastic has played a very important role in detecting and preventing the spread of the virus, while at the same time, there has been a significant increase in plastic waste. Plastics are usually polymers, i.e., materials that are formed by combining the same or similar molecules into a long chain. The basic raw materials used for polymer synthesis are crude oil, coal and natural gas. Many polymers are synthetic, i.e., artificially produced, but there are also polymers that occur naturally, such as cellulose, which is found in the cell walls of plants [1–4].

Plastics are often multi-layered and contain different types of plastics with different additives and adhesives. Special recycling processes are therefore required depending on the material. Due to their widespread use and economic viability, the production of plastics is constantly increasing. As plastic production increases, so does environmental pollution as a result of inadequate waste management.

Plastic pollution threatens the oceans, food safety and quality, human health, tourism (Figure 1) and affects global climate change. Discarded or poorly stored plastic in the biosphere is transported via precipitation and most often ends up in the seas and oceans. There, it breaks down into smaller and smaller pieces or fragments—microplastics. Microplastics pose a particular threat, especially when they enter the water supply, as they are

difficult to filter and remove. Apart from the fact that removing microplastics from water is a technically challenging and expensive process, once microplastics enter lakes and oceans, they are difficult to trace [5].

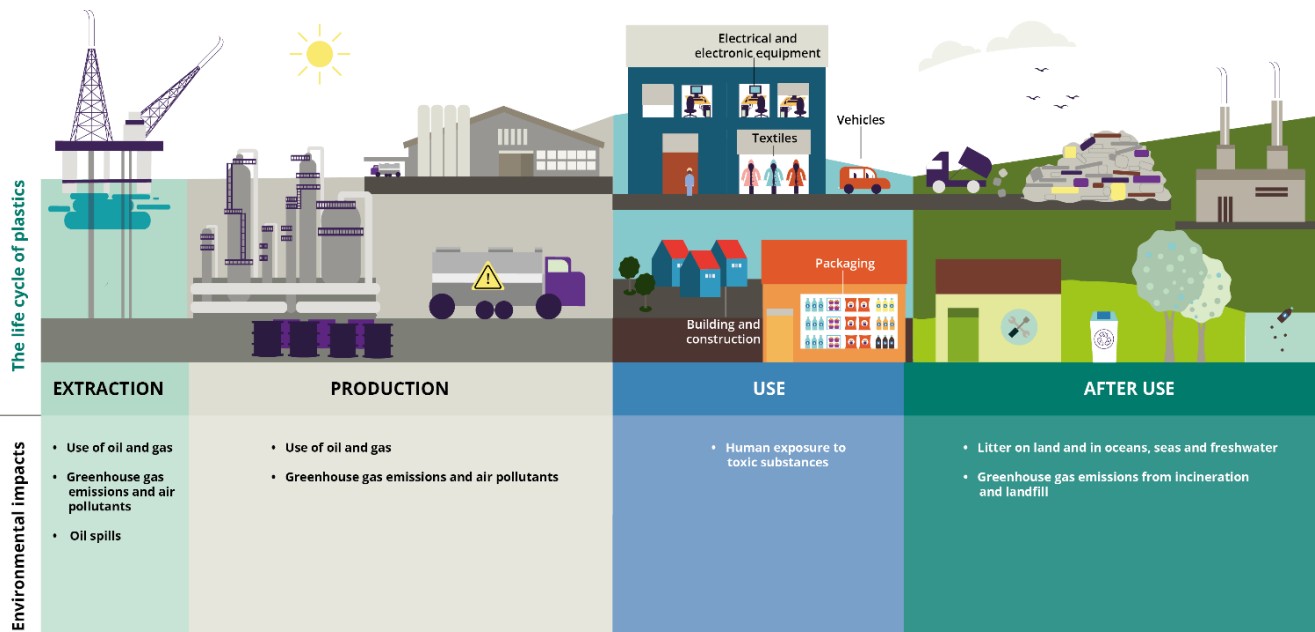

**Figure 1.** Environmental impact of plastic during its production and consumption [6].

As the concentration of microplastics in the environment increases, so does the likelihood that they will accumulate in food webs. Like macroplastics and macrofauna, microplastics can lead to the entanglement and physical disruption of organisms, and if ingested, can lead to gut engulfment and starvation in some organisms. In addition, plastics are composed of various chemicals, including polymers, dyes and plasticizers, most of which have toxic properties [7–11].

The chemical and physical properties of the shredded particles remain unchanged, meaning that microplastics have just as long a lifespan as does plastic. As plastic has become an integral part of people's lives, the possibility of microplastics being absorbed into the body has increased considerably. In fact, micropolastics are today so widespread that recent studies confirmed their presence in fetal aminochorial membranes [12]. Ingested microplastic particles can enter in the respiratory system and cause interstitial lung disease, chronic bronchitis and oxidative stress. If microplastics are ingested over a longer period of time, chromosomal changes, infertility, obesity and, in the latter case, the occurrence of tumors are possible [13].

The effects of plastic on organisms are complex and depend on the characteristics of the plastic object (chemical structure and additives, size, shape and age), the extent and type of exposure, and the species, life stage and characteristics of the organism. Environmental conditions can also play a major role in the responses of organisms and ecosystems to plastic, due to the joint effects of other stressors such as ocean acidification, global warming or increased pollution [14]. Given the lack of consistency in research on the effects of microplastics on different species, the lack of standardized experimental and research studies, and the multiple routes of exposure, we cannot say that we understand the long-term environmental or ecological effects of microplastics, especially in the context of multiple different stressors. Ongoing plastics research, especially if it is preventative (i.e., testing materials in development or testing environmental hazards under possible future pollution scenarios), will provide us with knowledge to prevent new problems from occurring [1].

Biodegradable bags were developed in response to plastic pollution and the large amounts of packaging waste generated worldwide. The diversity of biodegradable materials and their different properties make it difficult to make simple, generic assessments, such as saying that biodegradable products are all "good" or all petrochemical-based products are "bad". After the initial phase of pilot plants in the 1990s, the subsequent increase in the production of biodegradable (bio)plastics by small specialized and established companies has reached an industrial level since 2000, and a significant proportion of new biodegradable plastics are now of renewable origin. Depending on their origin, biodegradable polymers can be classified as either biologically or petrochemically based. Biodegradable polymers are mainly naturally biodegradable and are made from substances of natural origin (plants, animals or microorganisms), such as polysaccharides (e.g., starch, cellulose, lignin and chitin), proteins (e.g., gelatin, casein, wheat gluten, silk and wool) and lipids (e.g., plant oils and animal fats/vegetable oils and animal fats), while biodegradable polymers of petrochemical origin, such as aliphatic polyesters (e.g., polylactid acid (PLA) and polycaprolactone (PCL)), aromatic copolyesters (e.g., polyhydroxybutyrate (PHB)) and polyvinyl alcohol (PVA), are produced via synthesis from monomers obtained through petrochemical refining and have a certain degree of essential biodegradability.

Biodegradable bags are considered environmentally friendly products that are stable under different atmospheric conditions, that completely decompose into $CO_2$ and water in soil and that compost in an average of six weeks (ASTM standard D5338). The decomposition of the bags requires certain bacteria that are not present in the storage conditions at home, in a factory or in a classic warehouse, so the bags cannot begin to decompose or lose their strength during use. The most commonly used basic raw material is polyester, which is obtained from corn dextrose. The product is extremely elastic and has no negative effects on human health (ASTM standard D5338). To achieve overall benefits, biodegradable plastics must offer benefits to waste management systems in addition to their performance and cost, and could serve as a potential solution to overcrowded landfills. Many polymers on the market are designed to be degradable, meaning they break down into smaller pieces and can even break down into residues that are invisible to the human eye. Although it is assumed that the degradation products will eventually biodegrade, there are no data demonstrating complete biodegradability in a reasonably short period of time. Therefore, hydrophobic plastic waste with a large surface area can potentially enter the water and other parts of the ecosystem. The impact of biodegradable microplastics on the environment is almost completely unexplored [15–18].

The use of biodegradable polymers is now seen as an alternative and a way to reduce the growing problem of plastic pollution. However, recent studies shows that the degradability of these materials in the natural environments does not usually match the degradation times described in the technical standards for this type of bag. The standards on the basis of which the bags are classified as biodegradable are defined under certain conditions and in certain parameter ranges prescribed by the standard (microbial activity, salinity, pH, radiation, temperature, pressure, etc.), and represent only a small part of the potential conditions in the environment. In addition, all the environmental conditions mentioned vary considerably, which is not considered in standardized biodegradation tests. Therefore, numerous studies today indicate that the biodegradability of compostable polymers in the natural environment is overestimated based on the existing technical standards for awarding certificates for biodegradable plastics [19]. In addition, several studies have shown that bags labeled as compostable do not degrade quickly under environmental conditions and are potentially toxic to organisms due to the chemical additives used in their production [20,21], while Ribba et al. [22] have demonstrated the negative effects of biodegradable microplastics on numerous freshwater organisms.

The aim of this study was to analyze the physical–mechanical degradability of different types of declared-biodegradable polymer-based bags commonly available on the Croatian market in order to analyze their life cycle and the degradation process under different real environmental conditions, where these bags may end up.

## 2. Materials and Methods

Samples of selected biodegradable polymer bags and plastic polymer bags were exposed in soil, home compost, freshwater (in a lake) and the air. The selected environmental conditions correspond to the conditions under which commercially available single-use plastic polymer bags and/or biodegradable polymer bags are most likely to end up, given the disposal and management methods for this type of waste.

For the study, three types of biodegradable bags frequently available on the Croatian market were selected and the change in their physical and mechanical properties was compared with a commercially available plastic bag and controls (bags stored in room conditions). The biodegradable bags used were labeled as compostable bags according to the European Union standards [23]. In this paper, the bags mentioned are referred to as 'white' (label 7P0258; 10 L), 'green' (label 7P0012; 10 L), 'brown' (label KB 20 02 08; 30 L) and 'blue' (plastic bag) so as not to emphasize their identity, as they are commercially available (Supplement Figure S1).

### 2.1. Conditions and Methods of Polymer Bags Exposure in Environments

The investigation of the mechanical properties (degradation) of the four types of bags mentioned was carried out in a total of five different environments and environmental conditions. The exposure of bags in different environments took place simultaneously in the 6-week period (28 May 2021–9 July 2021) and were located within a radius of 5 km from Maksimir Park (Zagreb, Croatia). This area is characterized by a similar air temperature and weather conditions to those of the park, were gathered from the meteorological station located in Maksimir park (Supplement Table S1) and included data on daily temperatures, precipitation, average wind speed, and soil temperature at 2, 5, 10, 20 and 30 cm, as well as insolation duration gathered for the research period (May–July 2022). Water temperature, pH and dissolved oxygen were directly measured in the field, as described in Section 2.1.3. In total, 6 samples of polymer-based bags of each type (white, green, brown and blue) were exposed according to the procedures described in the following chapters. Out of the 6 bags, 3 samples of bags of each type were collected after three weeks of exposure, and the remaining 3 samples of bags of each type were collected six weeks after the start of exposure, which is considered the half-life of biodegradable bags (ASTM standard D5338). For some used bags, a period of 6 weeks is also explicitly stated as the period for their decomposition. As control groups, two samples of each type of bag were stored in room conditions.

### 2.1.1. Exposure in the Soil

Samples of each bag type were exposed at the experimental site of the Faculty of Agriculture in Maksimir. In total, 6 pieces of each bag type were placed in "cages" (dimensions $20 \times 50 \times 100$ cm) with a diameter of approximately 0.5 mm to protect the samples from larger insects and small rodents (Figure 2), in accordance with previous research [24]. Each sample was exposed at the same depth of 20 cm below ground level. Soil temperature at 20 cm was obtained from the local meteorological station located in Maksimir Park (Supplement Table S1).

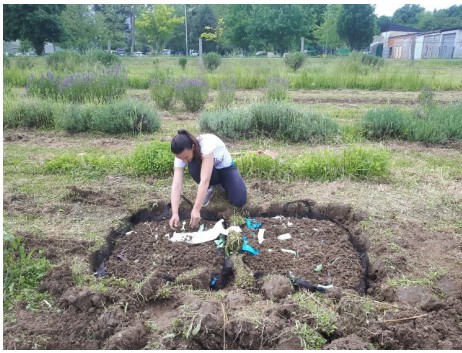

**Figure 2.** Placement of samples in the soil (source: Martina Fileš, 2021).

### 2.1.2. Exposure in the Compost

The samples of all bag types were placed in glass aquaria measuring $30 \times 30 \times 50$ cm, filled with compost and covered with a protective film (Figure 3). To create the conditions, i.e., to mimic the conditions of a composter, the aquariums with the samples were kept outdoors.

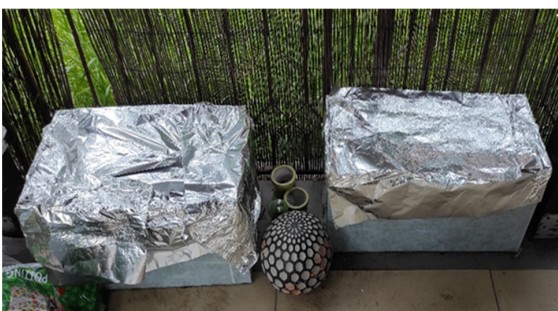

**Figure 3.** Placement of samples in the compost (source: Martina Fileš, 2021).

### 2.1.3. Exposure in the Freshwater Environment

Samples of each bag type were placed in the third lake in Maksimir, in net bags that were flooded by the weight of stones that were also placed in the net bags (Figure 4). The bags were placed in the lake in consultation with and under the supervision of employees of the public institution for the management of protected areas of the city of Zagreb. Since exposure was performed in the lake, no flow existed. Water temperature, pH and dissolved oxygen were measured 8 times during the experiment, using HI9813 probe (Hanna Instruments, Smithfield, VA, USA; Supplement Table S1).

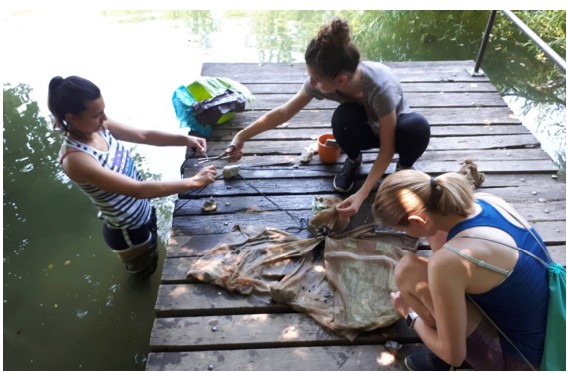

**Figure 4.** Placement of samples in the freshwater environment (source: Martina Fileš, 2021).

### 2.1.4. Exposure on the Air

The samples were suspended from a rope and exposed to atmospheric conditions in the air within a radius of 5 km around Maksimir Park, during the same period as the samples in other environments were suspended.

### 2.2. Measurement of Degradation Parameters—Morphological and Physical–Mechanical Properties of the Samples

The collected samples (3 samples of each type of polymer bag) were removed from the environment after 3 weeks and 6 weeks and taken to the laboratory for mechanical tests with standard atmosphere conditions, where various conditions were measured. The parameters of mechanical disintegration of the bags are described below (see Sections 2.2.1–2.2.3).

### 2.2.1. Analysis of Changes in Surface Structure and Sample Damage

In line with previous research [24,25], these parameters were observed visually and qualitatively via surface microscopy using a field emission scanning electron microscope

(FE-SEM Mira II LMU, Tescan, Czech Republic) at a magnification of 500× and 3000× to gain more detailed insights into surface changes and damage to the polymer material. The analysis of the morphological and surface changes in the tested biodegradable samples was carried out at the Department of Textile Chemistry and Ecology of the Faculty of Textile Technology at the University of Zagreb. Prior to analysis with the FE-SEM, the samples were vaporized with Cr in a QUORUM Q150 Test device for 2 min due to the non-conductivity of the polymer in order to obtain a clearly visible image of the surface. A green and a brown bag were selected for examination with the FE-SEM, as clearly visible changes could be seen on the brown bag after exposure to the environment, while no changes could be seen on the green bag (although it was labeled as a biodegradable bag) even after six weeks.

### 2.2.2. Tensile Breaking Force and Breaking Elongation

The tensile breaking force (F) and elongation at break ($\varepsilon$) were measured for each sample type and sampling in accordance with the ISO 527-1:2019 standard using TensoLab 3000 Strength Tester, which monitors the changes in the strength of each bag. In accordance with the standard, the samples were prepared with dimensions of 250 × 1000 mm, with 5 samples in the machine direction (MD) and 5 samples in the cross-direction (CD). The stretching speed was 300 mm/min in accordance with the specified ISO standard.

### 2.2.3. Mass Per Unit Area

The mass per unit area ($m_A$, g/m²) of the sample was determined on samples with a diameter of 10 cm² by determining the mass of each bag type on the control bag and the bag after exposure and calculating the mass per unit area, in accordance with the ISO 17554:2014 standard and previous studies [24].

### *2.3. Statistical Data Processing*

Differences in the measured parameters of the physical–mechanical properties (breaking force and elongation at break as well as mass per unit area) in the longitudinal (English machine direction, MD) and transverse (English cross-direction, CD) directions were investigated using non-parametric methods due to the small sample size (5 measurements per bag type in each environment). Comparisons between properties of different bag types exposed in different environments and respective controls (each bag type stored in room conditions) across exposure weeks were performed using the non-parametric Kruskal–Wallis ANOVA test and the post hoc Dunn test. In cases where a sample was missing (e.g., due to a high level of decomposition, a sample of white and brown bags after 6 weeks in the compost), a comparison between the control and exposed group (after 3 weeks of exposure) was performed using the Mann–Whitney U test.

Analyses of the differences in the measured mechanical parameters of bag decomposition over time and in different environments were performed using the Statistica software package (version 13.5.017. TIBCO Software Inc., Palo Alto, CA, USA). A significance level of 5% was used for all statistical analyses ($p < 0.05$).

## 3. Results

### *3.1. Micrograph Identification by FE-SEM*

SEM images (Figures 5 and 6) show the surfaces of the samples of biodegradable polymers before and after exposure to the tested environment (air, soil, compost and freshwater) for 3 and 6 weeks. During the microscopic analysis, the tests were performed at different magnifications to detect changes in the surface structure of the polymer, but magnifications of 500× and 3000× were chosen for the presentation of the results. When analyzing the green bag samples (Figure 5) exposed to different types of environments and comparing them with the initial sample, certain changes in the surface structure in the form of polymer fragments (residues) were detected. Obtained changes were more pronounced when the samples were exposed to the soil, and to a lesser extent when the samples were exposed to

air. Polymer samples exposed to compost and freshwater show a large amount of agglomerates on the surface of the polymer compared with the initial (referent) sample, which may be the result of environmental residues, but also the effect of the type of environment on the degradation of the polymer studied, depending on the exposure time.

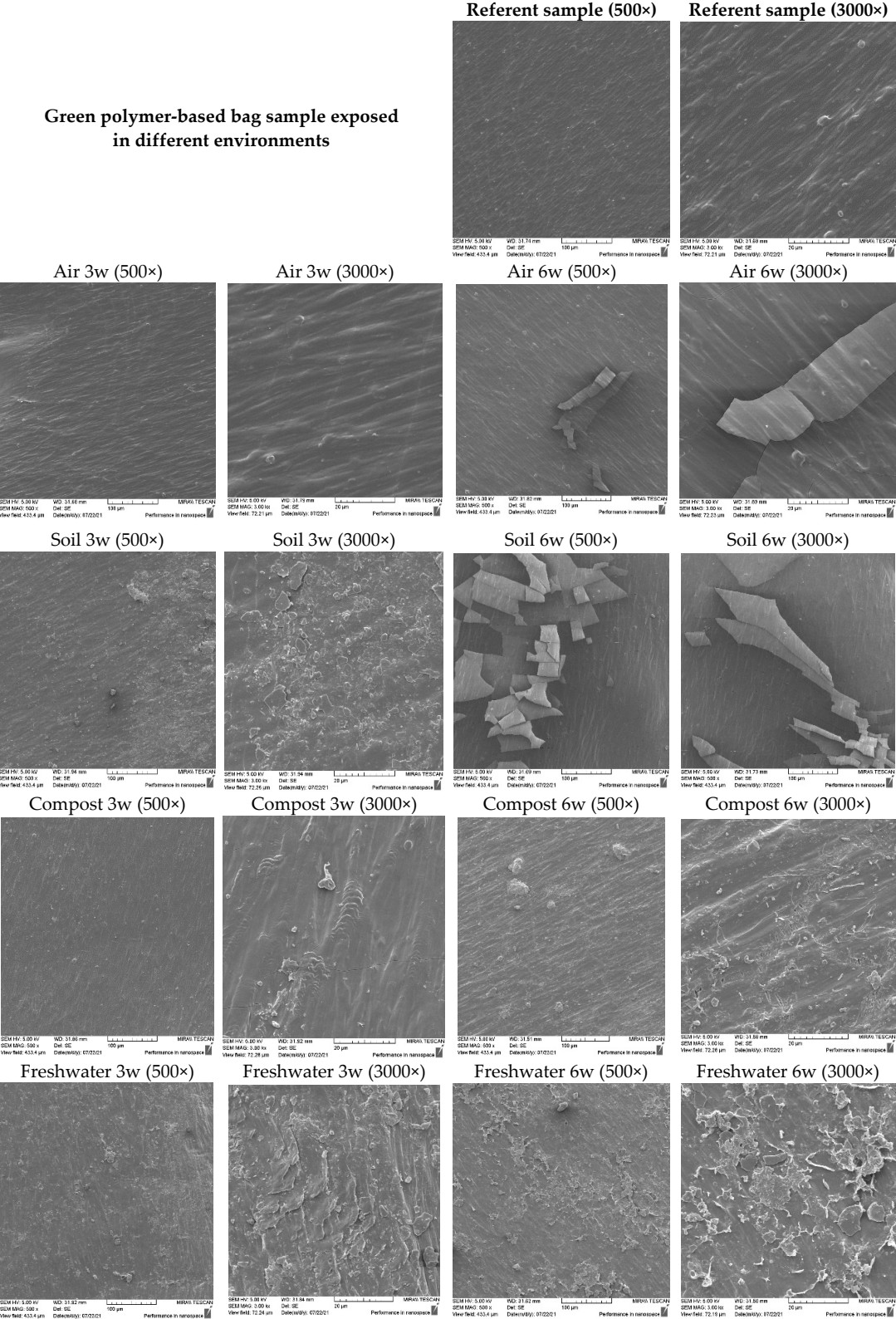

**Figure 5.** Green polymer-based bag sample in all environments after 3 (3w) and 6 (6w) weeks of exposure, imaged via FE-SEM microscopy, under magnifications of 500× and 3000×.

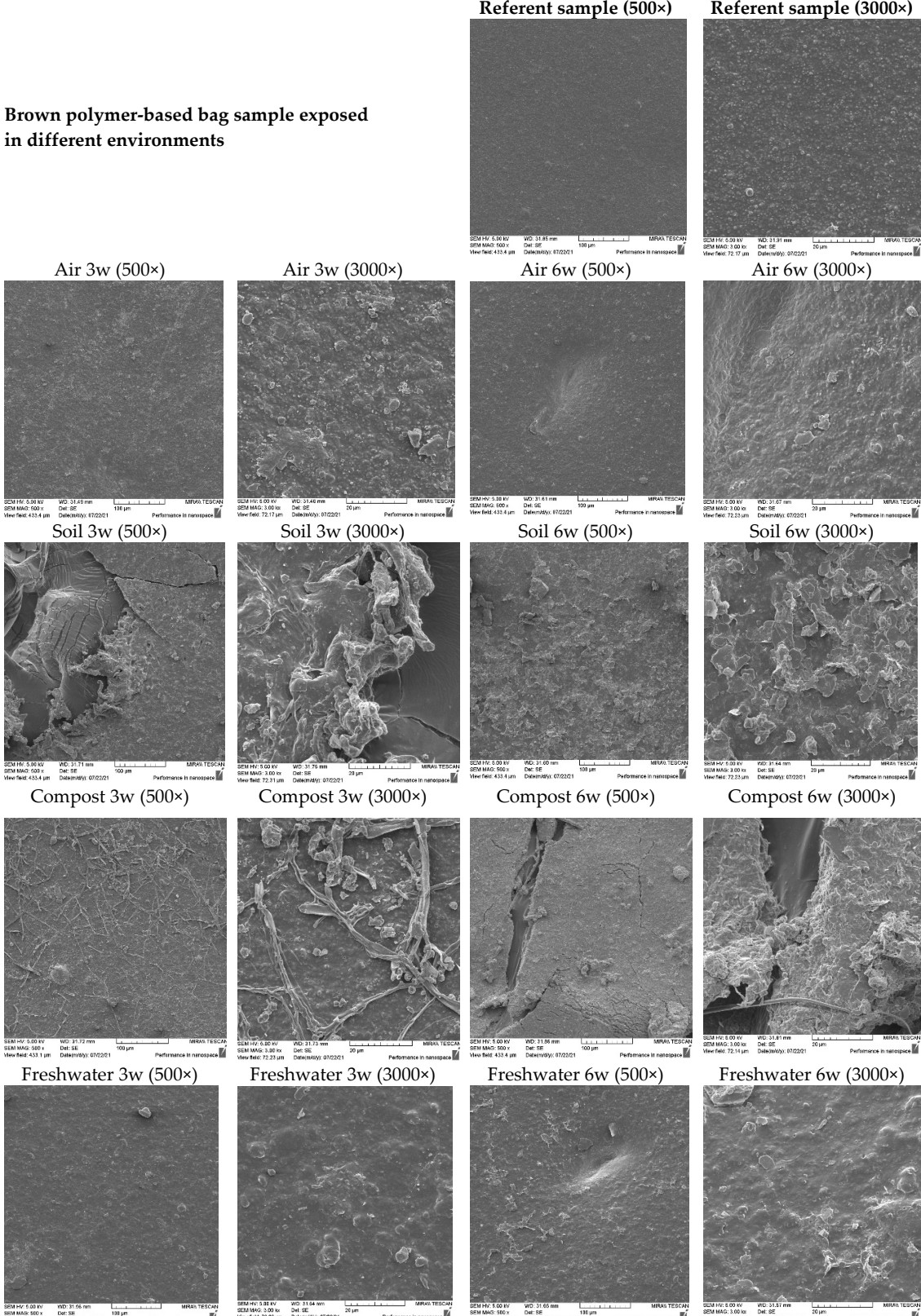

**Figure 6.** Brown polymer-based bag sample in all environments after 3 (3w) and 6 (6w) weeks of exposure, imaged via FE-SEM microscopy, under magnifications of 500× and 3000×.

The analysis of the SEM images of the brown bag samples (Figure 6) showed significant changes and damage to the structural surface of the polymer caused by different types of environment and exposure times compared with the initial sample. The largest changes and visible damage to the surface structure were observed in polymer samples exposed to compost and untreated soil after only 3 weeks of exposure, while the smallest visible damage in the form of agglomerates was observed on the surface of samples exposed to air and freshwater. Polymer samples exposed to the soil showed a greater amount of environmental residues (and/or agglomerates of degraded polymer), especially after 6 weeks of exposure. The obtained results of changes in the surface structure in the form of degradation are not in line with the results of the mechanical properties test, which proved the complete degradation of the tested polymer after exposure to the soil after 6 weeks. These differences may have been affected by the area of sample collection and the size of the sample taken for SEM analysis.

With regard to the type of environment, the duration of decomposition, the type of polymer and the change in mechanical properties (point 3.2), the SEM only gives a visual representation of the morphological changes in the surface due to exposure to different types of environments and for different periods of time (after 3 and 6 weeks of exposure). Therefore, the analysis of the resulting changes due to different types of environments should be considered from the point of view of all the properties tested.

### 3.2. Statistical Analysis of Physical–Mechanical Property Degradability of Tested Polymer-Based Bags under Environmental Conditions

The analysis of the differences in the measured mechanical properties revealed statistically significant differences between the different bag types in the machine direction (longitudinal, MD) and cross-direction (cross, CD) for the breaking force (F), and for the elongation at break ($\varepsilon$), as well as statistically significant differences in the mass per unit area between all bag polymer types (Table 1). Therefore, all bags were treated separately in the further analyses.

**Table 1.** Differences in the tensile breaking force (F, N) and elongation at break ($\varepsilon$, %) between the different polymer-based bag types in transverse (CD) and longitudinal (MD) directions, and mass per unit area ($m_A$, g/m$^2$), analyzed with the Kruskal–Wallis ANOVA test. Statistically significant differences are marked with (*).

| Properties | Direction | Df/N | H | *p* |
|:---:|:---:|:---:|:---:|:---:|
| F/N | CD | 3/20 | 14.874 | **0.0019 *** |
| F/N | MD | 3/20 | 8.600 | **0.0351 *** |
| $\varepsilon$/% | CD | 3/19 | 13.528 | **0.0036 *** |
| $\varepsilon$/% | MD | 3/20 | 16.590 | **0.0009 *** |
| $m_A$/g/m$^2$ | / | 3/40 | 34.780 | **0.0000 *** |

F—tensile breaking force (N); $\varepsilon$—elongation at break (%); $m_A$—mass per unit area (g/m$^2$); (Df/N—degrees of freedom/sample size; H—value of the Kruskal–Wallis test; *p*—*p*-value, significance level.

Similarly, for each polymer-based bag type, a significant difference was found in the measured parameters of breaking force and elongation at break between the different environmental types (Table 2); therefore, the changes in the reported mechanical properties for each bag type were also analyzed separately for each environment. The surface mass did not differ significantly between environments for any bag type (Table 2).

**Table 2.** Differences in tensile breaking force (F, N) and elongation at break ($\varepsilon$, %) between the different bag types in the transverse (CD) and longitudinal (MD) direction, and mass per unit area ($m_A$, g/m$^2$) between different type of environments for each individual type of bag, analyzed with the Kruskal–Wallis ANOVA test. Statistically significant differences are marked with (*).

| Type of Bags | Parameter | Direction | Df/N | H | *p* |
|---|---|---|---|---|---|
| White | F/ N | CD | 4/40 | 22.352 | **0.0002 *** |
| | F/ N | MD | 4/40 | 16.427 | **0.0025 *** |
| | $\varepsilon$, % | CD | 4/39 | 24.166 | **0.0001 *** |
| | $\varepsilon$, % | MD | 4/39 | 16.442 | **0.0025 *** |
| | $m_A$, g/m$^2$ | / | 4/9 | 5.133 | 0.2739 |
| Green | F/ N | CD | 4/55 | 26.618 | **0.0000 *** |
| | F/ N | MD | 4/54 | 9.828 | **0.0434 *** |
| | $\varepsilon$, % | CD | 4/54 | 26.822 | **0.0000 *** |
| | $\varepsilon$, % | MD | 4/54 | 15.007 | **0.0047 *** |
| | $m_A$, g/m$^2$ | / | 4/12 | 8.408 | 0.0777 |
| Brown | F/ N | CD | 4/45 | 31.535 | **0.0000 *** |
| | F/ N | MD | 4/45 | 33.962 | **0.0000 *** |
| | $\varepsilon$, % | CD | 4/45 | 31.535 | **0.0000 *** |
| | $\varepsilon$, % | MD | 4/45 | 29.705 | **0.0000 *** |
| | $m_A$, g/m$^2$ | / | 4/10 | 5.155 | 0.2718 |
| Blue | F/ N | CD | 4/50 | 26.745 | **0.0000 *** |
| | F/ N | MD | 4/50 | 3.606 | 0.4620 |
| | $\varepsilon$, % | CD | 4/50 | 11.576 | **0.0208 *** |
| | $\varepsilon$, % | MD | 4/50 | 4.240 | 0.3744 |
| | $m_A$, g/m$^2$ | / | 4/10 | 5.864 | 0.2096 |

F—tensile breaking force (N); $\varepsilon$—elongation at break (%); $m_A$—mass per unit area (g/m$^2$); (Df/N—degrees of freedom/sample size; H—value of the Kruskal–Wallis test; *p*—*p*-value, significance level.

### 3.2.1. Comparison of Changes in the Mechanical Properties of Different Types of Polymer-Based Bags during Six-Week Exposure in Different Environments

White Polymer-Based Bag

Exposure of the white polymer-based bag to air and freshwater did not result in significant mechanical changes in tensile strength or elongation (Kruskal–Wallis ANOVA and Mann–Whitney U-test, $p < 0.05$). Exposure to soil resulted in a significant decrease in tensile breaking force in the longitudinal direction (F:MD; H $_{(2,20)}$ = 6.253, $p = 0.0439$) and a significant decrease in elongation in the longitudinal direction (E:MD—H $_{(2,19)}$ = 8.335 $p = 0.0155$) and in the transverse direction (E:CD; H $_{(2,19)}$ = 8.053, $p = 0.0178$), with a significant difference between the control and exposure conditions after 6 weeks in most cases (Figure 7A–C). Exposure to compost resulted in a decrease in elongation in the transverse direction (E:CD—Z = 2.327, $p = 0.02$; Figure 7D), with decomposition after 6 weeks being so severe that it was not possible to take a sample of the bag required for testing the specified parameters in accordance with the method described.

Green Polymer-Based Bag

Placing the green polymer-based bag in compost did not result in any significant mechanical changes in tensile breaking strength or elongation (Kruskal–Wallis ANOVA, $p < 0.05$). Contrary to expectations, the green bag showed a statistically significant increase in tensile breaking strength in the longitudinal direction during exposure to air (H$_{(2,15)}$ = 6.980, $p = 0.0305$; (Figure 8A), while exposure to water and soil resulted in a statistically significant decrease in tensile breaking strength and elongation in the transverse direction (F:CD soil—H $_{(2,25)}$ = 6.803, $p = 0.0333$; E:CD soil—H $_{(2,25)}$ = 11.187, $p = 0.0037$; F:CD freshwater—H $_{(2,15)}$ = 11.180, $p = 0.0037$; E:CD freshwater—H $_{(2,15)}$ = 11.180 $p = 0.0037$, Figure 8B–E)).

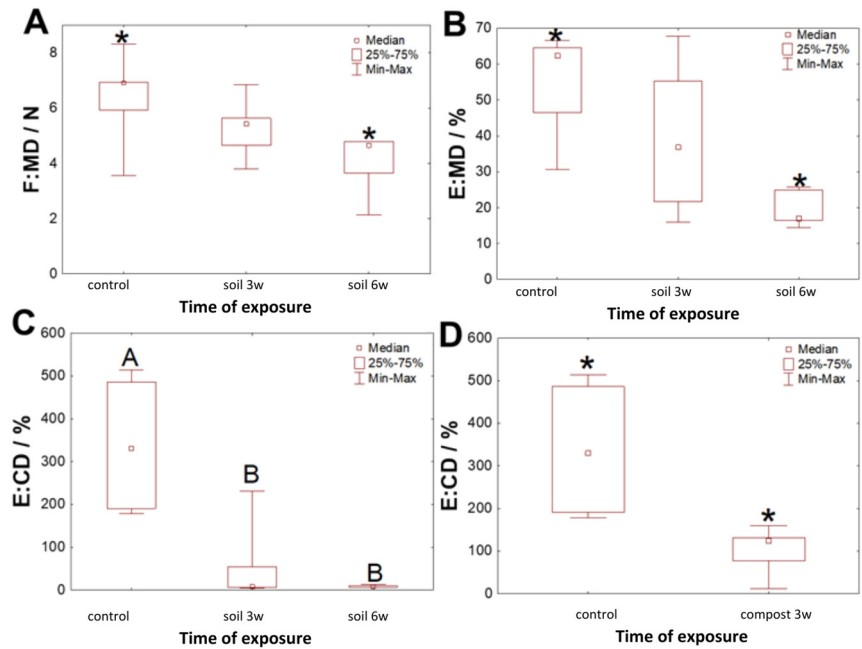

**Figure 7.** Differences in (**A**) tensile breaking force in the longitudinal direction (MD) and elongation in (**B**) the longitudinal direction (MD) (**C,D**) and transverse direction (CD) during exposure to soil and to compost. Statistically significant differences between the groups are marked with an asterisk (*) or different letters.

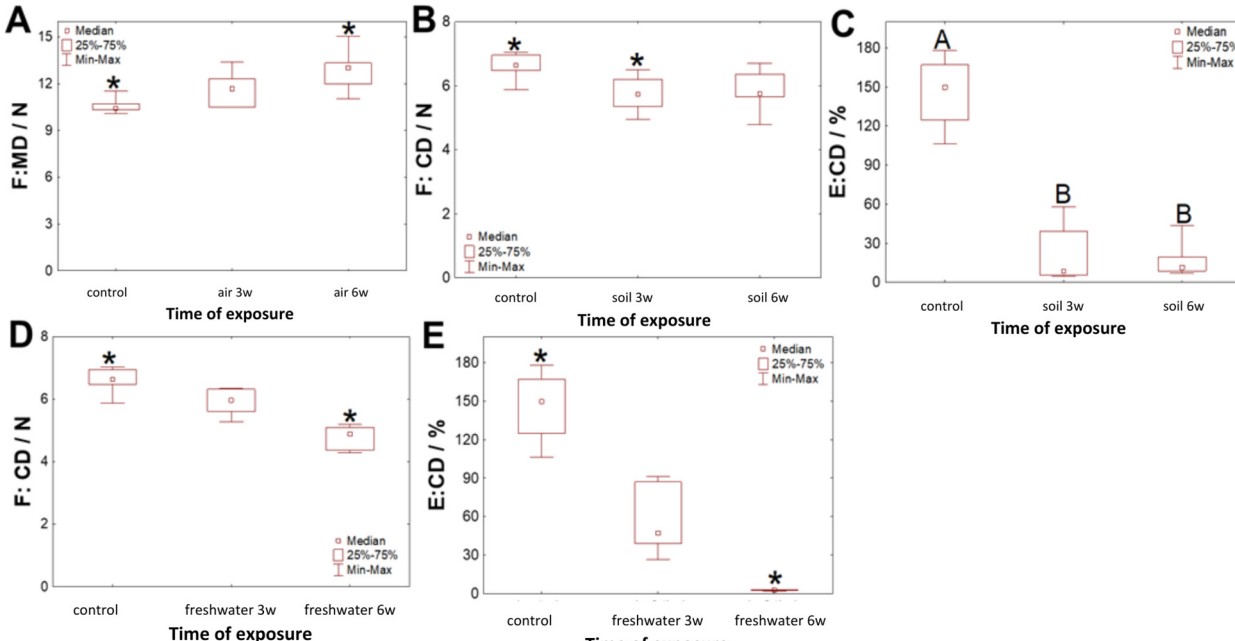

**Figure 8.** Differences in (**A**) tensile breaking force in the longitudinal direction (MD) in air; (**B,C**) tensile breaking force and elongation during exposure to soil and freshwater (**D,E**). Statistically significant differences between the groups are marked with an asterisk (*) or different letters.

Brown Polymer-Based Bag

Exposure of the brown polymer-based bags resulted in a statistically significant reduction in tensile breaking strength and elongation in at least one direction (longitudinal and/or transverse) in all environmental types (Figures 9–12). Exposure in soil resulted in a significant reduction in tensile breaking strength and elongation in the transverse

and longitudinal directions (Kruskal–Wallis ANOVA, $p < 0.05$ in all cases), with a significant difference recorded between the control group and the groups exposed for three and six weeks (Figure 9A–D). Exposure to air resulted in a significant decrease in tensile strength and elongation in the transverse direction (F:CD—H $_{(2,15)}$ = 10.82, $p$ = 0.0087; E:CD—H $_{(2,15)}$ = 9.572, $p$ = 0.0083) after three- and six-week exposure (Figure 10A,B), while elongation was also statistically significantly reduced in the longitudinal direction after six weeks (E:CD—H $_{(2,15)}$ = 9.076, $p$ = 0.0107; Figure 10C). Exposure of the brown bags to freshwater resulted in a significant decrease in tensile breaking strength and elongation in the transverse direction (F:CD—H $_{(2,15)}$ = 9.5, $p$ = 0.0045; E:CD—H $_{(2,15)}$ = 9.62, $p$ = 0.0081) after 6 weeks of exposure (Figure 11A,B). Exposure to compost resulted in very severe decomposition of the bags after 6 weeks of exposure; therefore, it was not possible to collect adequate samples of the bags required for testing as described in Section 2.2.2). Exposure to compost resulted in a significant decrease in tensile strength and elongation in the transverse and longitudinal directions (Figure 12; Mann–Whitney U-test, N1 = N2 = 5; $p < 0.05$ in all cases).

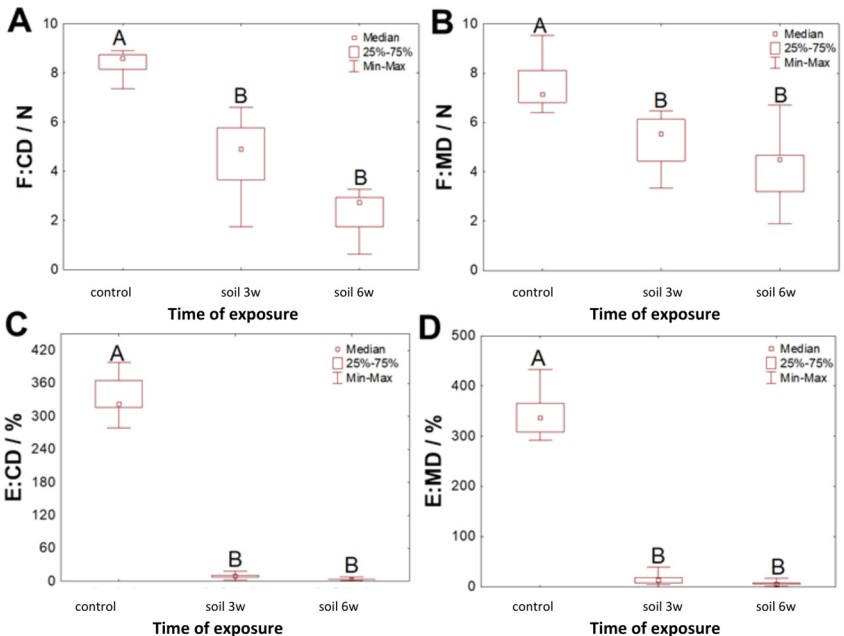

**Figure 9.** Differences in (**A**) tensile breaking force in the transverse direction (CD) and (**B**) in the longitudinal direction (MD); (**C**) elongation in the transverse direction (CD) and (**D**) longitudinal direction (MD) during exposure of the brown bag to soil. Statistically significant differences between the groups are marked with different letters.

Blue Polymer-Based Bag—Conventional Plastic Bag

Exposure of the blue polymer-based bag, together with the green polymer-based bag, resulted in the smallest changes in the measured parameters in the different environmental types. Exposure to air resulted in no statistically significant changes in tensile breaking strength and elongation in any direction, while exposure to soil resulted in a statistically significant decrease in elongation in the transverse dimension after 6 weeks (Figure 13A, E:CD—H $_{(2,25)}$ = 6.778, $p$ = 0.0337). When exposed to freshwater, a statistically significant difference in tensile strength in the transverse direction was observed between the control bag and the sample exposed for 6 weeks (Z = 2.507, $p$ = 0.0012, N1 = N2 = 5; Figure 13B), while the sample exposed for 3 weeks was missing due to the loss of part of the bags in water. When exposed to compost, a statistically significant decrease in tensile strength in the transverse direction was observed in the bags exposed to compost for 3 and 6 weeks compared with that of the control bags (Figure 13C).

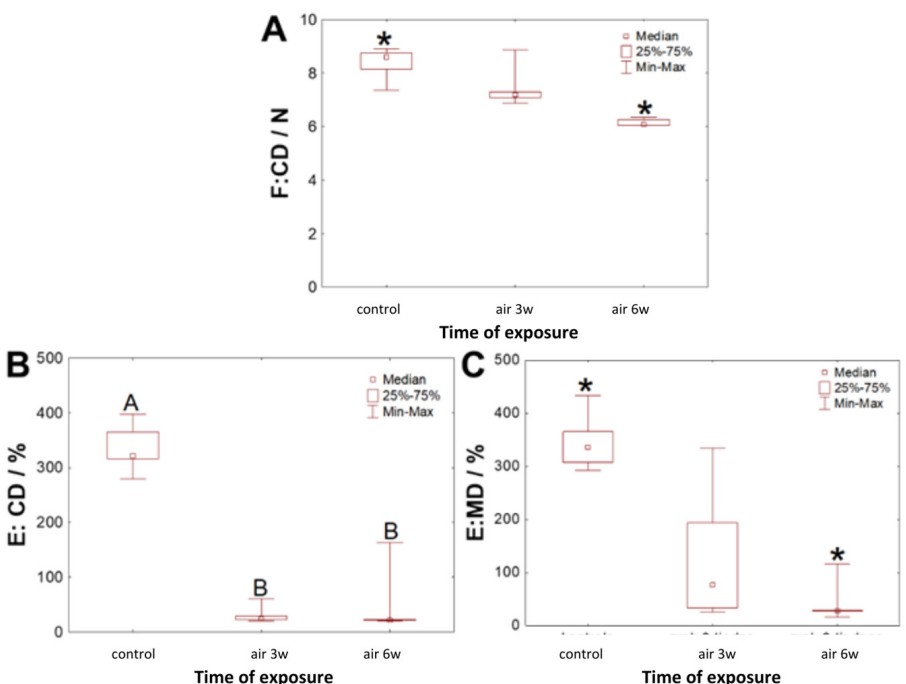

**Figure 10.** Differences in (**A**) tensile breaking force in the transverse direction (CD); elongation (**B**) in the transverse direction (CD) and (**C**) longitudinal direction (MD) during exposure of brown bag on air. Statistically significant differences between the groups are marked with an asterisk (*) and different letters.

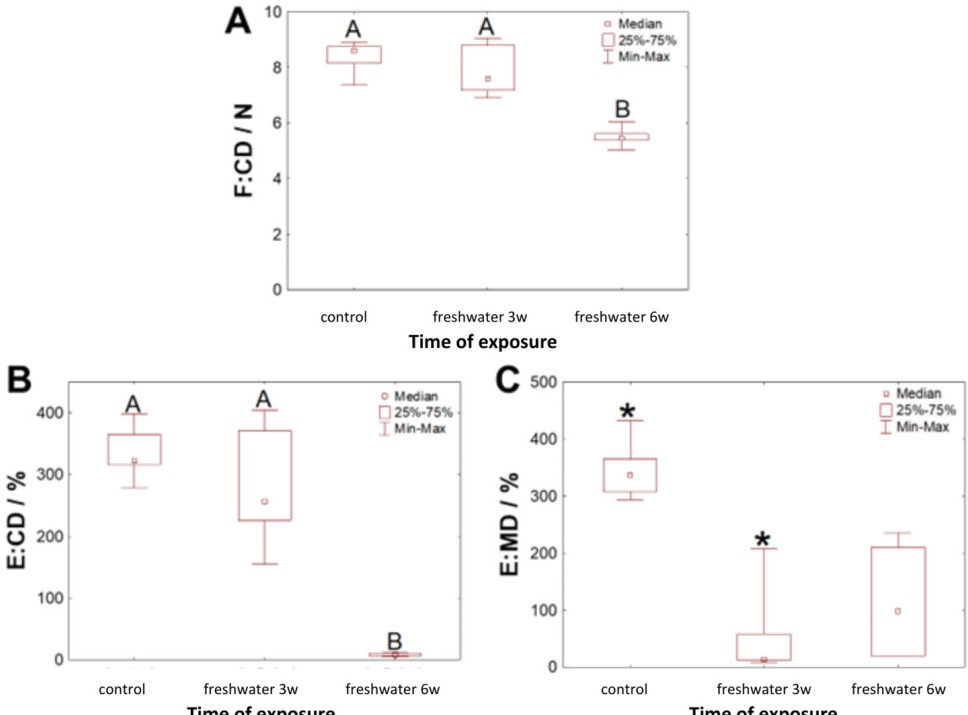

**Figure 11.** Differences in (**A**) tensile breaking force in the transverse direction (CD); elongation (**B**) in the transverse direction (CD) and (**C**) longitudinal direction (MD) during exposure of the brown bag in freshwater. Statistically significant differences between the groups are marked with an asterisk (*) and different letters.

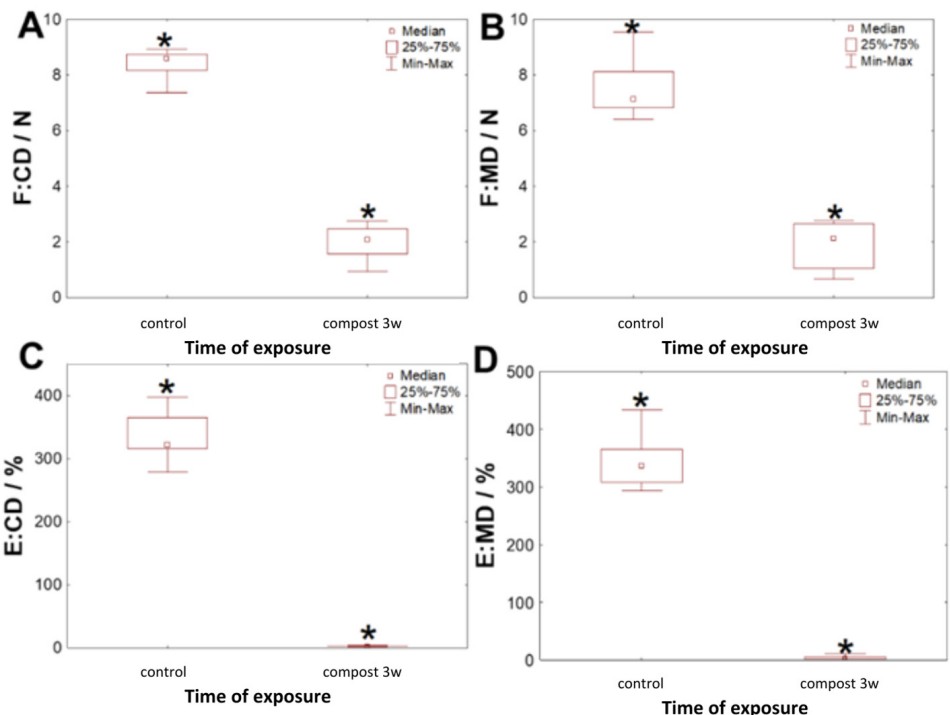

**Figure 12.** Differences in (**A**) tensile breaking force in the transverse direction (CD) and (**B**) in the longitudinal direction (MD), and in elongation (**C**) in the transverse direction (CD) and (**D**) in longitudinal direction during exposure of the brown bag to compost. Statistically significant differences between the groups are marked with an asterisk (*).

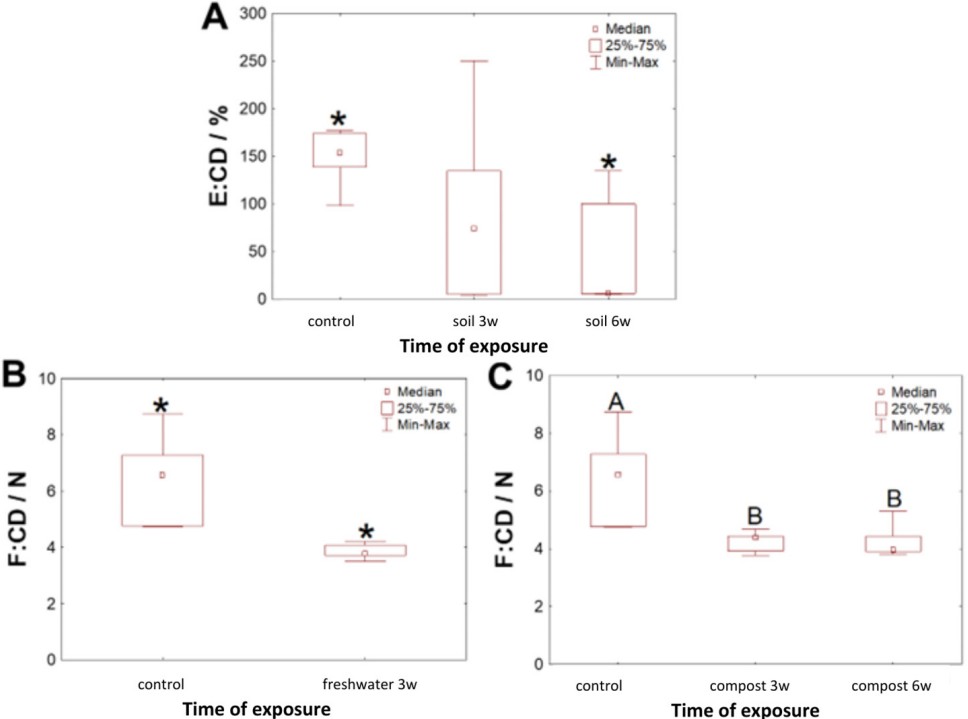

**Figure 13.** Differences in (**A**) elongation in the transverse direction (CD) during exposure of the blue bag to soil; tensile breaking force in the transverse direction (CD) during exposure to (**B**) freshwater and (**C**) compost. Statistically significant differences between the groups are marked with an asterisk (*) and different letters.

## 4. Discussion

In this study, the degradation (change in mechanical properties) of polymer-based bags declared as compostable and biodegradable was investigated in different types of environments where these bags may "unintentionally" end up (air, soil and freshwater) as well as in compost (where these polymer-based bags were expected to end up). The change in mechanical properties of the biodegradable polymer-based bags was compared with a commercially available plastic polymer-based bag. Although the changes in mechanical properties and visual changes/damage to the surface texture observed were more pronounced in the biodegradable polymer-based bags compared with the conventional plastic polymer-based bags, none of the tested polymer-based bag types completely degraded during the six-week trial period. The three and six-week trial periods were chosen based on the available information; the six-week period was indicated as the degradation time for some of the polymer-based bags tested, since this is the stated degradation time of biodegradable polymer-based bags according to ASTM standard D5338. A significant change in mechanical properties was observed in all types of biodegradable polymer-based bags in the specified time, including a smaller change in the conventional plastic polymer-based bag (the blue one). Of the biodegradable polymer-based bags, the brown bag showed the best degradation and the greatest change in mechanical properties, followed by the white bag, and the green bag showed the least change. The green bag was not defragmented upon visual inspection, even in the compost where the brown and white bags were almost completely defragmented, but it showed visual signs of damage when exposed to water. Similar results were obtained by Napper and Thompson [24], who examined bags for their composition, including biodegradable bags, oxo-biodegradable bags (conventional plastic with additives to mimic biodegradation) and compostable bags, with the oxo-biodegradable bag showing the least degradation. Although we tried to find out the exact composition of the bags, we did not receive an answer from the manufacturer, so we can only assume that the results obtained here depend on the raw material composition of the polymer-based bags. In contrast to the other biodegradable bags, the green polymer-based bag showed the highest similarity of changes in mechanical properties to those of the blue commercial plastic bag, and visual inspection did not reveal any pronounced damage to the surface structure, as seen in the brown and white bags.

Depending on the different types of environment in which the bags were exposed, the greatest changes in mechanical properties and the greatest damage to the surface structure were observed on the bags exposed in compost, although with some types of bags (green), the mentioned changes were noticeable only after 6 weeks of exposure. In this period, the degradation of other types of polymer-based bags (brown and white) was so great that an analysis of changes in mechanical properties was not possible after 6 weeks. After recording the changes in the compost, the greatest changes in mechanical properties were recorded in soil. In other types of environments (in water and in air), no significant changes in mechanical properties or significant structural changes were recorded. As for the decomposition of the bags in the compost, the obtained results were expected, as these bags are designed to be compostable and under assumption that in most cases they end up in landfills and will not represent an additional environmental problem. Additionally, in biological waste landfills they can also end up processed and reused as nutrients for plants, for example. In the research by Klauss and Bidlingmaier [26], it was shown that composts containing biodegradable polymer-based bags did not show differences in quality parameters compared with conventional compost consisting only of green waste and had the same positive effect on the soil and the plant. The problem arises when the bags end up in natural environment, such as water and air. For example, Napper and Thomson showed that decomposition in sea water was almost non-existent even after several years of exposure and that not a single bag, regardless of the length of exposure, completely decomposed [24]. Similarly, a study by Artru and Lecerf [27] showed that biodegradable bags decomposed at a much slower rate than those labeled

on the bags, based on certification standards, indicating their resistance to degradation in natural environments.

In our study, not a single polymer-based bag type was completely degraded after 6 weeks of exposure, and the least change in morphological (surface) structure and mechanical properties was observed in air and then in freshwater. It can therefore be assumed that biodegradable polymer-based bags, when released into the aquatic environment or into the air, degrade slowly and pose the same hazard in the aquatic environment as do conventional plastic bags, as also suggested by other authors [27]. Even though the decomposition of the polymer-based bags was visible in the soil and compost and the mechanical properties changed significantly, other research results indicate that the biodegradable polymer PLA (polylactic acid, the most commonly used biodegradable plastic) generates more microplastics than does the conventional biodegradable polymer PE (polyethylene) [28]. It therefore remains to be investigated whether the use of biodegradable bags will lead to an increase in plastic pollution in the long term.

## 5. Conclusions

From the research conducted, it appears that various types of commercially available biodegradable polymer-based bags in the Republic of Croatia show signs of fragmentation and changes in mechanical properties during short-term exposure to the environment, and we assume that the (chemical) composition of the raw material has the greatest influence on the possibility of their decomposition. Different types of environmental conditions influenced the mechanical decomposition of the bags either very successfully (compost; soil) or almost negligibly (air; freshwater).

It can be concluded that the bags showing the best potential for decomposition are white and brown polymer-based bags disposed of in compost. Of all the polymer-based bag types tested, the brown bags showed better degradation potential in different environments (compost and soil, with the weakening of mechanical properties by water).

The best decomposition of all types of polymer-based bags was achieved in compost, followed by soil, and the worst was achieved in water and air. This shows us that the decomposition of these bags, if they enter the aquatic environment (freshwater or the sea), will take a very long time, which may have undesirable consequences for the organisms that live in such environments, similar to the case for conventional plastic bags.

From all this, it can be concluded that biodegradable polymer-based bags must also be disposed of in prescribed landfills and that further studies must show whether or not complete decomposition takes place, in what period of time and for which particles.

**Supplementary Materials:** The following supporting information can be downloaded at https://www.mdpi.com/article/10.3390/su16062579/s1: Figure S1: Polymer-based bags before degradation in different environments; Table S1: Meteorological data.

**Author Contributions:** Conceptualization, S.E.R. and S.H.; methodology, S.E.R. and S.H.; software, M.F. and S.H.; formal analysis, M.F. and A.L.; investigation, M.F. and A.L.; resources, M.F.; writing—original draft preparation, S.E.R.; writing—review and editing, S.H., M.F. and A.L.; visualization, M.F.; project administration, A.L.; funding acquisition, S.E.R. All authors have read and agreed to the published version of the manuscript.

**Funding:** This research received no external funding.

**Institutional Review Board Statement:** Not applicable.

**Informed Consent Statement:** Not applicable.

**Data Availability Statement:** Data are contained within the article and Supplementary Materials.

**Acknowledgments:** Special thanks go to Zorana Kovačević for conducting the analysis of morphological and structural surface changes in the tested biodegradable samples using the FE-SEM at the Department of Textile Chemistry and Ecology, University of Zagreb Faculty of Textile Technology.

**Conflicts of Interest:** The authors declare no conflicts of interest.

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
