# Peer review of "Analysis of the Mechanical Degradability of Biodegradable Polymer-Based Bags in Different Environments"

_sustainability, doi:10.3390/su16062579_

Round 1
Reviewer 1 Report
Comments and Suggestions for Authors
In this manuscript, the authors provided a systematic decomposition study of four types of plastic bags in different environments. The degradation behavior of each group of samples was further analyzed mainly by observing the surface micromorphology and physical-mechanical properties. This is of practical significance for assessing the biodegradation potential of waste plastic bags in the environment and their potential impact on the environment and organisms. However, there are still several necessary corrections that need to be made in the manuscript.
1) It is recommended to organize the discussion about different types of plastics in lines 43-79 into a table;
2) The last paragraph of Introduction is too long. It is recommended to split it into several paragraphs to enhance the logic and readability of the text;
3) The last part of Introduction should briefly summarize the research content of this paper and establish a logical connection with the previously raised research questions;
4) The full name of abbreviations such as PLA, PCL, and PHB should be appended when they appear for the first time;
5) Specific composition information of the four samples in the study should be provided, which will add to the completeness and novelty of the research content of the manuscript;
6) It is recommended to add comparative figures of the macroscopic morphology of plastic bags before and after degradation for each group of samples, so that the impact of different degradation environments on the samples can be more intuitively examined;
7) Regarding the observation results of the surface micromorphology of samples after degradation in different environments, what morphological characteristics on the SEM images in Figure 6 are the authors based on to judge the degree of degradation? What are the criteria for judging the possible residues indicated by the red circles? Because substances with similar morphological characteristics are also seen in other SEM images, but they are not circled, which is a bit confusing. Please provide further explanation;
8) There is an overlapping error in the annotations in Figure 7 and needs to be corrected;
9) Why are there only SEM test results of the micromorphology of the "green" and "brown" samples, but what about the other two groups of samples?
10) Does the soil environment in physical-mechanical performance analysis refer to untreated or treated soil? No corresponding instructions and annotations were found;
11) Lines 203, 211 and 218, "Figure x" should be supplemented with a specific serial numberï¼›
Comments on the Quality of English LanguageThe English language needs to be checked, and there are some grammatical errors.
Author Response
Dear Reviewer,
The authors carefully read all questions and suggestions from the reviewers and provided answers to all questions. Since some questions were very similar, the answers were summarised in one document.
The authors would like to thank the reviewers for their contribution and all the suggestions that have been incorporated into the revised version of the paper.
Yours sincerely!
Please find the answers in attachment.

Reviewer 2 Report
Comments and Suggestions for Authors
The manuscript titled "Degradability Analysis of Biodegradable Polymer-Based Bags in Different Environments" analyzed decomposition of three types of biodegradable and one type of plastic bags in different environment. The experimental design and data presentation needs to be clearly stated and thus requires a major revision before publication. Please find the comments below:
1. Line 183: The authors referred mechanical properties to evaluate degradation of bags, but degradation can also be correlated with physicochemical properties. Please discuss the rationality to only evaluate mechanical properties.
2. Line 200: Please describe the dimension of bags applied in this study.
3. Line 203/211/218: Please double check and number the figures.
4. The experimental design of 4 environments are way too simplified. It is not clearly stated or controlled of pH, temperature, or flow in the water, or the air temperature and air flow etc, and the variance can impact the degradation process. Please add more elaborations to specify the environment of how the studies were conducted.
5. Can the authors comments on the salt concentration impact the degradation of bags? for example in a sea water environment?
6. Can the authors elaborate how the time points were designed? Why 3 and 6 weeks were investigated but no time zero point?
7. The mass weigh of bags can be a direct indication of degradation and have the authors conducted any studies on that?
9. Can the authors elaborate why only green and brown bags were studied by SEM?
10. Figure 12-13: Is there 1-panel graph missing? Or not applicable?
11. There are only 29 references and more current studies in the are are needed to engage with the recent scholars. Please include a few references from Edgar group of modified Cellulose as they develop potential biodegradable materials for sustainability:
"Regioselective chlorination of cellulose esters by methanesulfonyl chloride"
"Efficient synthesis of glycosaminoglycan analogs"
Author Response

(The authors gave the same response as above.)

Reviewer 3 Report
Comments and Suggestions for Authors
This is a relatively good and interesting paper with regard to the biodegradation of plastic bags investigated in different conditions. The authors selected 3 types of commercially available biodegradable plastic bags and one conventional plastic bag for comparison, tested in sel-compost, soil, fresh water and air for 6 weeks. They observed clear degradation of plastic bags but not a complete degradation for any sample tested. This finding is very useful for undersanding the real biodegradability of biodegradable plastci bags. It can be recommended for publication but some revisions should be made.
1. Introduction is too long, it can be concise significantly if considering the readers are professional in this filed;
2. "white", "brown", "green" and "blue" are used to indicate the four types of bags tested but without releasing the indentity of samples. However, the components of plastic bags strongly influence the biodegradability under different envionments, if possible, some rough information of these bags should be provided for the readers;
3. The test was carried out in a specific area for 6 weeks, the average ground temperature should be provided, which is helpful for better understanding.
4. The biodegradation of plastic bags are believed to be mainly dependent on the bacterial activity in the environment. Since the test was done in fresh water for 6 weeks. it is inappropriate to draw a simple conclusion that the bags can not easily degrade in marine conditions.
5. If possible, it is recommended to give some discussions or hints why the degradation behavior of a single plastic bag is quite different under different environments, i.e. compost, soil, water and air.
Comments on the Quality of English Languageit is good
Author Response

(The authors gave the same response as above.)

Round 2
Reviewer 1 Report
Comments and Suggestions for Authors
After considering the changes made by the Authors, I would like to recommend this revised manuscript for publication.
Reviewer 2 Report
Comments and Suggestions for Authors
The manuscript has been revised accordingly based on the reviewer's comments. The clarity and quality has been improved, therefore I would recommend its publication in Sustainability.